# S-Nitrosylation of Tissue Transglutaminase in Modulating Glycolysis, Oxidative Stress, and Inflammatory Responses in Normal and Indoxyl-Sulfate-Induced Endothelial Cells

**DOI:** 10.3390/ijms241310935

**Published:** 2023-06-30

**Authors:** Cheng-Jui Lin, Chun Yu Chiu, En-Chih Liao, Chih-Jen Wu, Ching-Hu Chung, Charles S. Greenberg, Thung-S. Lai

**Affiliations:** 1Department of Medicine, MacKay Medical College, New Taipei 25245, Taiwan; lincj@mmh.org.tw (C.-J.L.); enchih@mmc.edu.tw (E.-C.L.); u632@mmc.edu.tw (C.-J.W.); chchung@mmc.edu.tw (C.-H.C.); 2MacKay Junior College of Medicine, Nursing and Management, New Taipei 25245, Taiwan; 3Division of Nephrology, Department of Internal Medicine, MacKay Memorial Hospital, New Taipei 25245, Taiwan; 4Institute of Biomedical Sciences, MacKay Medical College, New Taipei 25245, Taiwan; p01485-630@mmc.edu.tw; 5Division of Hematology/Oncology, Medical University of South Carolina, Charleston, SC 29425, USA; greenbec@musc.edu

**Keywords:** tissue transglutaminase, TG2, TGase, transamidation activity, nitric oxide, NO, indoxyl sulfate, IS, reactive oxygen species, ROS, post-translational modification, PTM, endothelial NO synthase, eNOS, glyceraldehyde 3-phosphate dehydrogenase, GAPDH, NADPH oxidase, NOX

## Abstract

Circulating uremic toxin indoxyl sulfate (IS), endothelial cell (EC) dysfunction, and decreased nitric oxide (NO) bioavailability are found in chronic kidney disease patients. NO nitrosylates/denitrosylates a specific protein’s cysteine residue(s), forming S-nitrosothios (SNOs), and the decreased NO bioavailability could interfere with NO-mediated signaling events. We were interested in investigating the underlying mechanism(s) of the reduced NO and how it would regulate the S-nitrosylation of tissue transglutaminase (TG2) and its substrates on glycolytic, redox and inflammatory responses in normal and IS-induced EC injury. TG2, a therapeutic target for fibrosis, has a Ca^2+^-dependent transamidase (TGase) that is modulated by S-nitrosylation. We found IS increased oxidative stress, reduced NADPH and GSH levels, and uncoupled eNOS to generate NO. Immunoblot analysis demonstrated the upregulation of an angiotensin-converting enzyme (ACE) and significant downregulation of the beneficial ACE2 isoform that could contribute to oxidative stress in IS-induced injury. An in situ TGase assay demonstrated IS-activated TG2/TGase aminylated eNOS, NFkB, IkBα, PKM2, G6PD, GAPDH, and fibronectin (FN), leading to caspases activation. Except for FN, TGase substrates were all differentially S-nitrosylated either with or without IS but were denitrosylated in the presence of a specific, irreversible TG2/TGase inhibitor ZDON, suggesting ZDON-bound TG2 was not effectively transnitrosylating to TG2/TGase substrates. The data suggest novel roles of TG2 in the aminylation of its substrates and could also potentially function as a Cys-to-Cys S-nitrosylase to exert NO’s bioactivity to its substrates and modulate glycolysis, redox, and inflammation in normal and IS-induced EC injury.

## 1. Introduction

Circulating uremic toxins, including indoxyl sulfate (IS), are present in chronic kidney disease (CKD) patients [1]. Even after hemodialysis, IS remains in circulation in CKD patients, and the levels vary between 7–343 μM with an average of 120–140 μM in serum [1,2]. Cardiovascular disease (CVD) occurs early, and declining renal function is associated with cardiovascular morbidity and mortality in CKD patients [3]. The predominant vascular pathologies in CKD patients are endothelial cell (EC) dysfunction and vascular calcification, both of which are caused by increased oxidative stress and reduced NO bioavailability [3]. The angiotensin-converting enzyme (ACE) inhibitor and angiotensin II (Ang II) type I receptor (AT1R) blocker are part of therapeutics for combating CKD progression, as circulating Ang II is reported to promote EC dysfunction and hypertension as well as extracellular matrix (ECM) dysregulation [3]. However, the regulation of ACE and its ACE2 isoform in IS-induced EC is largely unknown. Moreover, EC is in the innermost layer of blood vessels in direct contact with the systemic effects of IS, but the molecular events mediating NO bioavailability are largely unknown.

Physiological levels of NO derived from endothelial NO synthase (eNOS) are beneficial and a key regulator of vascular tone and proper EC function [3]. NO exerts its protecting and signaling effects through the generation of cGMP and the S-nitrosylation/denitrosylation of numerous EC proteins [4,5]. The eNOS activity is tightly regulated by the available substrate L-arginine, the co-factors tetrahydrobiopterin (BH4), Ca^2+^ and NADPH, phosphorylation, S-nitrosylation, and interactions with other proteins [6,7]. In addition, eNOS needs to be a dimer for the production of NO [6]. The stability of the dimeric form (coupling) is maintained by optimal concentrations of the substrate and co-factors. In contrast, the uncoupled (monomeric) form of eNOS is unable to produce NO and generates O_2_^−^ instead [6]. Phosphorylation of human eNOS Ser^1177^ or Thr^495^ results in an increase or decrease of activity [6]. The production of NO during eNOS uncoupling reacting with O_2_^−^ can form cytotoxic peroxynitrite (ONOO^−^), which can contribute to the oxidation of BH_4_ [6]. S-nitrosylation is considered a feedback mechanism to regulate eNOS activity when its concentrations are high [6]. The molecular basis contributing to decreasing NO bioavailability in IS-induced endothelial injury is unknown. The S-nitrosylated proteins and the functional consequences of these post-translational modification(s) (PTMs) have never been investigated under normal conditions and in IS-induced EC injury.

Once generated, NO is extremely susceptible to redox, resulting in S-nitrosylation of target proteins with NO’s bioactivity [7,8]. The cellular homeostasis of SNOs is regulated by two main mechanisms, transnitrosylation, and denitrosylation, which cooperatively control the steady-state cellular levels of SNOs [7,8]. Transnitrosylation is the reversible transfer of an NO group from one SNO carrier (known as an S-nitrosylase) to another, thereby allowing for the effective propagation of NO’s bioactivity [7,8]. GSH/GSSG plays an important role in NO’s physiology and pathology. GSH binds NO (form GSNO) and serves as the abundant physiological SNO carrier and delivers NO to potential protein thiols [5]. Transnitrosylation can also occur between a S-nitrosylase and the target proteins (Cys-to-Cys transfer) in close proximity with the right conformation [5]. We previously demonstrated that an EC-rich tissue transglutaminase (TG2) is robustly S-nitrosylated in vitro and in vivo, and SNO-TG2 is functioning as a NO carrier on the EC surface to inhibit neutrophils adhesion [9,10]. We postulated that TG2 could function as an S-nitrosylase to transfer SNO to its substrates [9,10]. To do this, TG2 must be in close proximity to eNOS (or other types of S-nitrosylase) to transmit NO’s bioactivity. As SNOs can regulate a protein’s structure, function, and signaling, the results could be used as an early biomarker to limit further inflammation during IS-induced injury.

TG2 is a multifunctional protein with a high affinity for GTP involved in G-protein signaling and other enzymatic functions, including transamidase (TGase), that are modulated by NO and redox [11,12,13,14]. The well-established TGase catalyzes an irreversible Ca^2+^-dependent cross-linking reaction between either a Q-containing protein and a K-containing protein or a Q-containing protein and a free primary amine such as putrescine etc. (also referred as the aminylation reaction) resulting in the PTM of a target protein’s structure and function. The TGase is inhibited by GTP but can be partially reversed by the transient rise of Ca^2+^ [12,15].

Under physiological conditions, intracellular TG2/TGase is in a GTP-bound (known as “compact or close”) inactive conformation and could be activated by the rise of Ca^2+^, while extracellular TG2 can be TGase active or inactive [9,12]. In an extracellular environment, Ca^2+^ is in the mM range, and Ca^2+^ -bound TG2/TGase is in an “open active or extended” conformation but was found to be inactive due to disulfide formation under oxidative stress [16]. TG2 has 18 Cys thiols, several of which are polynitrosylated and could dispense part of SNOs upon adding ~ mM of Ca^2+^ in vitro [9,10]. In this study, we employed a specific, irreversible peptide mimetic inhibitor of TG2/TGase, ZDON, reported to form a substrate-bound “open” conformation to investigate the potential function of TGase and as an S-nitrosylase. Decreased NO bioavailability and increased Ca^2+^ during cellular injury could contribute to abnormal activation of TG2/TGase and TG2′s ability to function as an S-nitrosylase. EC constitutively expresses rich levels of TG2 [17]. TG2 is situated in or at the cytoplasm, nucleus, and plasma membrane [12,15]. TG2 is also secreted onto EC surfaces, where it binds the co-receptor fibronectin (FN) and into the extracellular matrix (ECM) [12,15]. TG2 is, therefore, a potential candidate to function as an S-nitrosylase to dispense NO to target proteins and modulate EC functions, but this hypothesis requires further investigation. Current studies aim to understand the molecular basis of reduced NO bioavailability and the function of TG2 when functioning as a TGase and an S-nitrosylase in normal and in IS-induced EC injury.

TG2 is a therapeutic target for CKD, contributing to renal fibrosis and vascular calcification [18,19]. TG2/TGase is also involved in the assembly and activation of NADPH oxidase 2 (GP91/Phox, NOX2), one of the major pathways in generating intracellular ROS in ECs [20]. TG2 was also implicated in the dimerization of AT1R, the receptor for Ang II [21]. Using an in vivo CKD model, two different irreversible inhibitors of TG2 prevented a decline in kidney function and reduced the development of glomerulosclerosis and tubulointerstitial fibrosis by up to 77% and 92%, respectively [18,19]. The data demonstrated that inhibition of TG2/TGase activity offers a potential therapeutic option for CKD. SNO-TG2 is found in a young aorta but not in an aged aorta [22], and decreased SNO-TG is linked to an increase in TG2/TGase activity and vascular stiffness in eNOS^_^/^_^ mice demonstrating that TG2’s activation in vivo is modulated by eNOS [23]. Decreased NO bioavailability could lead to activation of TG2/TGase and could contribute to the development of fibrosis in CKD but has never been investigated in EC under normal and IS-induced injury.

Glycolysis is one major pathway to modulate oxidative stress. ECs utilize glycolysis to generate > 85% of their ATP in oxygen-replete conditions in a way similar to cancer cells [8]. Multiple glycolytic enzymes, including pyruvate kinase 2 (PKM2), glucose-6-phosphate dehydrogenase (G6PD), and glyceraldehyde-3-phosphate dehydrogenase (GAPDH), are known targets for ROS and/or S-nitrosylation [8]. ROS inhibits multiple glycolytic enzymes, including GAPDH and PKM2, to promote glucose substrate flux into the pentose phosphate pathway (PPP) [8,24]. ECs express PKM2 exclusively over PKM1, catalyzing one of the final steps of glycolysis, and PKM2 is also interacting with eNOS [24,25]. Nitrosylated PKM2 increases substrate flux through the PPP to generate NADPH (and GSH) to protect against oxidative stress [24]. Glucose-6-phosphate dehydrogenase (G6PD) catalyzes the rate-limiting step in the PPP to generate NADPH to generate NADPH [26]. S-nitrosylation of G6PD may serve to protect GAPDH from oxidant inactivation and to regulate glycolysis. How these enzymes/co-factors are modulated by NO in IS-induced endothelial injury is unknown. More understanding of the S-nitrosylation of these glycolytic enzymes is necessary to develop effective interventions for IS-mediated EC injury.

## 2. Results

To establish human umbilical vascular endothelial cells (HUVEC) as the EC model to investigate the effects of IS on decreased NO bioavailability, we first demonstrated that ROS generation was induced by IS treatment (Appendix A), as previously reported [3]. To investigate the dosage response in the initial phase of IS-induced injury, we treated cells with clinically relevant (0 to 250 μM) doses of IS for 0 to 6 h, and cell viability (determined by intracellular ATP levels) was measured. As shown in Figure 1, ATP levels were decreased in a time and IS-dosage-dependent manner (Figure 1A). Different caspases were found to be significantly activated at 250 μM, and cells died in a caspase-1-independent manner (Figure 1C). To investigate cell’s redox responses, S-nitrosylation, and inflammatory proteins associated with the initial phase of IS-induced injury, we selected 100 μM of IS and 4 h of incubation for data throughout this study at which cells were ~90% viable, and cells were still attached to the Petri dishes (Figure 1A and Appendix A).

A specific, irreversible inhibitor of TG2/TGase, ZDON (a membrane permeable active site peptide mimetic) [27], was employed to investigate whether it could attenuate the oxidative stress and inflammatory responses in IS-induced injury. Several irreversible inhibitors have been developed targeting the TG2/TGase active site Cys^277^ among which ZDON is widely studied, but its efficacy in EC has never been reported [27]. With an IC_50_ of 20 nM vs. TG2/TGase, ZDON was found to be selective on TG2 over other types of TGs, including TG_1_ (7.3 μM), TG_3_ (0.2 μM), and factor XIIIa (67 μM) [27]. The concentration ranges of ZDON used here were based on earlier studies that did not affect cell viability [27]. In the absence of IS, we found ZDON was able to improve ATP levels in a concentration-dependent manner and was increased to ~155% when 40 μM was used (Figure 1B). In the presence of 100 μM of IS, ZDON also improved ATP levels in a concentration-dependent manner and was increased to ~125% when ≥ 20 μM of ZDON was used (Figure 1B).

To investigate whether IS-mediated cell injury was related to caspase 1-mediated activation, we used a caspase-Glo^®^ 1 Inflammasome Luminescent Assay (Promega, Madison, WI, USA). In this assay, Z-WEHD-aminoluciferin, the luminescent substrate of caspase 1, was used either in the presence or absence of a specific caspase 1 inhibitor, Ac-YVAD-CHO, to determine whether other types of caspases were also activated (Figure 1C). As shown in Figure 1C, similar IS dosage-dependent response curves were observed either in the absence or presence of Ac-YVAD-CHO, indicating other types of caspases were also activated during IS-induced injury. This was a very sensitive assay, and we observed caspase activation at IS ≥ 125 μM. Caspases were found to be activated significantly higher at 250 μM of IS when compared with the vehicle control. When comparing with the same IS concentrations, ZDON (at 40 μM) slightly decreased caspase activation (Figure 1C).

### 2.1. Intracellular Cellular NADP^+^/NADPH and GSH/GSSG Ratios

The NADP^+^/NADPH and GSH/GSSG ratios are important indexes reflecting the cellular redox status [6]. GSH and NADPH are co-factors required for superoxide dismutase (SOD), GSH peroxidase, GSH reductase, and/or catalase for the conversion of O_2_^−^/H_2_O_2_ to H_2_O [6]. In the absence of ZDON, the NADP^+^/NADPH ratios were increased in an IS-dosage-dependent manner and were found to increase ~2-fold at 250 μM of IS when compared with the vehicle control (Figure 2A). Pretreatment with ZDON was able to reduce NADP^+^/NADPH ratios at all the IS concentrations, with the ratios only increasing ~1.4-fold at 250 μM of IS (Figure 2A). At all the IS concentrations, the NADP^+^/NADPH ratios were lower in the presence of ZDON, indicating an increase in NADPH generation (Figure 2B).

In the absence of ZDON, the GSH/GSSG ratios were decreased in an IS-dosage-dependent manner and were reduced by ~4-fold at 250 μM of IS (Figure 2B). In the presence of ZDON, the GSH/GSSG ratios were not significantly changed at IS ≤ 125 μM but were reduced by ~3-fold at 250 μM of IS (Figure 2B). At all IS concentrations, the GSH/GSSG ratios were slightly higher in the presence of ZDON, indicating an increase in GSH generation (Figure 2B).

### 2.2. Effect of ZDON on Extracellular Acidification Rate (ECAR) upon IS Exposure

Protection against ROS largely relies on the reductive power of NAPDH, whose levels are determined by glucose substrate flux into PPP. To evaluate the effects of IS on glucose substrate flux, we performed real-time monitoring of glycolytic metabolism using an extracellular flux analyzer and a glycolysis stress test kit (Agilent Inc., Santa Clara, CA, USA).

As shown in Figure 3, there was a biphasic effect of IS on glucose substrate flux. At 20 and 40 μM of IS, the ECAR was increased by ~30–40% but was decreased to a level similar to vehicle control when cells were treated with 80 μM of IS. At 20 and 40 μM of IS, the glycolytic capacity was increased by ~2-fold but was not significantly changed in the presence of 80 μM of IS, and the glycolytic reserve followed a similar trend as glycolytic capacity (Figure 3A). Based on the data on cell viability (Figure 1A) and preliminary studies performed in the absence of IS, ZDON was able to increase ECAR. We then investigated whether pretreatment of cells with ZDON before incubating with 80 μM of IS, we found ECAR was increased by ~20–30%, and the glycolytic reserve was also increased by ~2-fold (Figure 3B). The data indicated pretreatment with ZDON increased the glycolytic pathway to generate ATP.

### 2.3. Effects of ZDON on the Expression of eNOS and Phospho-eNOS upon IS Exposure

Upon exposure to 100 μM of IS, the expression of eNOS was induced 2-fold, and the levels were not significantly changed comparing cells previously treated with ZDON (Figure 4A). The expression of phospho-eNOS (phosphorylation at 1177 residue; designated as P^1177^-eNOS) was not significantly changed when treated with IS either in the absence or presence of ZDON (Figure 4A). When the protein band intensities were quantified using Image J and the data were presented as the P^1177^-eNOS/eNOS ratios. As shown in Figure 4B, IS reduced the P^1177^-NOS/eNOS ratio by 60%, and the ratio was further reduced by 2-fold when cells were pre-treated with ZDON.

### 2.4. In Situ TGase Assay to Identify Potential TG2/TGase Substrates

We next investigated whether in situ TG2/TGase activity was activated in cells treated with IS (Figure 5). Biotinylated pentylamine (BP), the primary amine substrate of TG2, was used to label TG2/TGase Q-substrate(s), a reaction also referred to as an aminylation reaction as described in [12,28,29]. In response to a rise in intracellular Ca^+2^, BP was shown to be aminylated to the potential Q-containing protein(s) by TG2/TGase [29].

When BP-labeled total lysates prepared from cells pre-treated with ZDON and with either 0, 100, or 200 μM of IS were incubated with streptavidin-conjugated beads to pull-down biotin-labeled substrates, we identified transcription factors, including NFkB and extracellular proteins fibronectin (FN), eNOS, GP91-Phox (NOX2), and IkBα that were serving as in situ TG2/TGase substrates (Figure 5A). These TGase-substrate band intensities were higher at 200 μM of IS when compared with either the vehicle control or 100 μM of IS (Figure 5A). As shown in Figure 5A, pretreatment with ZDON significantly reduced the BP-labeled substrates, indicating ZDON was acting against the activation of TG2/TGase and with specificity (Figure 5A). Several glycolytic enzymes, including G6PD, PKM2, and GAPDH, and cytoskeletal proteins β-actin were also labeled by BP, but the band’s intensity appeared to be similar across all the concentrations (Figure 5B), indicating that they were constitutively aminylated. The data demonstrated that TG2/TGase was activated in situ and played a role during IS-induced ECs.

### 2.5. SNO-RAC on Purified TG2, Total SNO Proteins and S-nitrosylation of TG2/TGase Substrate Proteins during IS-Induced Injury

We first investigated whether we could detect nitrosylation on a purified TG2 and see whether there was any difference in the nitrosylation of TG2 in the presence or absence of either 0.1 mM Ca^+2^ and/or ZDON (Figure 6A). During in vitro nitrosylation, ZDON was co-incubated with Ca^+2^ as it was reported that Ca^+2^ was required for ZDON to bind to the TG2 [30]. Nitrosylation was performed in vitro using a CysNO donor, and we could detect the nitrosylation of TG2 in the absence or presence of 0.1 mM Ca^+2^ as well as in 0.1 mM Ca^+2^/ZDON. The band intensity of TG2 was higher when the assay was performed in the absence of Ca^+2^ and Ca^+2^/ZDON (Figure 6A).

To investigate proteins modified by SNO, we then determined the amount of total SNO proteins and S-nitrosylation of TG2/TGase substrate proteins using SNO-RAC and immunoblot assays as described [31] (Figure 6). Following the SNO-RAC assay, the amount of total SNO proteins eluted from thiopropyl-sepharose beads was quantified by a BCA protein assay (Figure 6B). As shown in Figure 6B, in the absence of ZDON, total SNO proteins treated with 100 μM of IS were decreased by ~37% when compared with the vehicle control (Figure 6B). In the absence of IS, total SNO proteins did not change significantly either in the presence or absence of ZDON (Figure 6B). In the presence of 100 μM of IS, total SNO proteins were increased by ~60% when cells were pre-treated with ZDON (Figure 6B).

As shown in Figure 6C,D, the total SNO proteins eluted from thiopropyl-sepharose were analyzed by immunoblot, and each protein bands intensities were detected using specific antibodies against the identified TG2/TGase substrates (in Figure 5) and quantified using Image J. In the absence of ZDON, the levels of S-nitrosylated TG2, NF*k*B, IkBa, G6PD, and PKM2 treated with 100 μM of IS, were found to be higher than the vehicle control, while that of eNOS, GAPDH, and b-actin were slightly reduced (Figure 6C,D). In the presence of ZDON, the S-nitrosylated protein levels of eNOS, IkBa, NFkB, and GAPDH were increased, while the levels of TG2, PKM2, G6PD, and actin were not significantly changed in the presence of IS (Figure 6C,D).

### 2.6. Time Courses Studies on the Expression of TG2/TGase Substrates, Redox, and Inflammatory Proteins upon IS Exposure Either in the Presence or Absence of ZDON

To understand the expression levels of TG2/TGase substrate proteins upon being treated with IS either in the presence or absence of ZDON, total lysates were isolated from cells treated with 100 μM of IS for 0, 2, 4, and 6 h. The time course was selected based on Figure 1A data showing ~90% of cells were viable and was considered as the initial phase IS-induced injury. Special interest was focused on TG2/TGase substrates related to glycolytic enzymes, redox, inflammatory and extracellular proteins, and the data are shown in separate figures as up- or down-regulated (Figure 7).

As shown in Figure 7A,B, upon exposure to IS for 4 h, heme oxygenase 1 (HO1) was dramatically up-regulated by ≥ 22-fold either in the absence or presence of ZDON. Initial proteomic analysis and western blot validation indicated ACE was up-regulated by ~1.5-fold for 6 h either in the absence or presence of ZDON, and the expression of NFkB (P60) followed the expression patterns with a similar trend. In addition, GP91-Phox (Nox2) and G6PD were found to be slightly up-regulated. IS treatment induced the expression of Nox2 but not Nox1 and Nox4.

As shown in Figure 7C,D, the ECM protein fibronectin (FN) was found to be down-regulated dramatically by ≥10 or 6-fold, either in the absence or presence of ZDON for 6 h, respectively. The isoform ACE2 was downregulated by ≥2 or ≥6-fold, either in the absence or presence of ZDON for 6 h, respectively. The glycolytic enzyme, PKM2, was found to be downregulated by ~2 or ~1.5 fold, either in the absence or presence of ZDON for 6 h, respectively. In addition, the inflammatory protein IkBα was slightly downregulated either in the absence or presence of ZDON. All expression patterns showed in a time-dependent manner.

## 3. Discussion

This study represents the first detailed study on the molecular basis for decreased NO bioavailability and to understand the potential role of TG2 at the initial phase of IS-induced EC injury. In this model, we demonstrated that the reduced NO could be due in part to oxidative stress and the decreased eNOS co-factors, including NADPH, as well as the phosphorylation and S-nitrosylation of eNOS (Figure 2 and Figure 4). The increase in ROS generation could contribute to the oxidation of BH_4_ and the breakdown of NO (Appendix A). ZDON was used to probe the contribution of TG2/TGase and as an S-nitrosylase to transmit NO signaling. We also provided data on normal cellular response to IS, either in the absence or presence of ZDON. This is the first study on the efficacy of ZDON in modulating glycolysis, redox, and inflammation in normal and IS-induced ECs.

In the absence of ZDON, we first investigated the time course of ROS responses in IS-induced injury. Using DCFH-DA and JC-1 green as the fluorescence probes, we found that IS induced intracellular ROS and damaged mitochondria (Appendix A). We found GP91-Phoxwas induced by IS was the likely source of ROS (Figure 7A). The upstream regulator of ROS, NF*k*B, was also induced (Figure 7A). As the cellular response to oxidative stress, heme oxygenase I (HO1), the downstream target of the antioxidant response regulator of nuclear factor erythroid-2-related factor 2 (Nrf2), was dramatically up-regulated (Figure 7A). The cellular anti-oxidative system was overwhelmed by the continued increase in ROS and was consistent with the data showing that IS decreased the GSH and NADPH levels (Figure 2A,B). For GSH regeneration and ROS clearance, glucose substrate flux was shifted toward PPP to generate NADPH and was consistent with the immunoblot showing increased expression of G6PD while decreasing PKM2 expression in response to IS (additional discussion below) (Figure 7A). However, the increased NADP^+^/NADPH ratios upon IS exposure demonstrated cells were not generating sufficient NADPH levels despite increased expression of G6PD (Figure 2A and Figure 7A), suggesting that there were additional mechanisms in regulating the homeostasis.

We found increased expression of ACE upon IS exposure (Figure 7A). ACE converts circulating Ang I to Ang II (Renin-Angiotensin system; RAS) and also degrades bradykinin, an NO inducer, and is responsible for an increase in ROS, vasoconstriction, inflammation, and fibrosis as well as ECM dysregulation representing current targets in designing therapeutic to combat CKD progression [32,33,34]. Ang II is also reported to up-regulate NF*k*B, NOX, and ROS in kidney tubular cells treated with IS [35]. Moreover, IS dramatically reduced the expression of ACE2, a beneficial isoform of ACE that proteolyzes Ang II to Ang I [32,33,34] as early as 2 h (Figure 7C). Ang I counteracts the effects of Ang II through a MAS receptor and, thus, protects against organ injury [32,33,34]. Extracellular protein CCN1 (also called Cyr61, Cysteine-rich angiogenic inducer 61), reported to be up-regulated by Ang II, is an angiogenic-immediate early gene involved in the development and progression of arteriosclerosis [36]. CCN1 also leads to MMP1 activation and FN degradation [37]. FN levels were found to be dramatically reduced as early as 2 h and can be used as an early marker for IS-induced EC injury (Figure 7C).

Under oxidative stress, oxidation of BH_4_ could result in the uncoupling of eNOS [6]. In addition, eNOS’s activity is also regulated by phosphorylation and S-nitrosylation [6]. Although the expression of eNOS was slightly induced by IS treatment, the levels of P-eNOS, which up-regulate eNOS’s activity, were not increased (Figure 4A,B). The levels of SNO-eNOS, which inhibit eNOS’s activity, were mildly reduced by IS treatment (Figure 6C,D). The data indicated that eNOS was not in optimal conditions to generate NO under IS-induced injury. The data further supported that total SNO proteins were reduced by ~37% upon treatment with IS, while this decrease was reduced to <10% in the presence of ZDON, and the underlying mechanism(s) warrant further investigation (Figure 6B).

Studies reveal that the S-nitrosylation of ~3000 target proteins is the major pathway exerting NO’s bioactivity [38]. NO produced inside the cell is quickly reacting with NO-carriers (or S-nitrosylases) proximal to the NO source (i.e., eNOS) [39]. To date, there are less than 10 proteins identified as S-nitrosylases, including GAPDH [38]. Up to individual proteins, SNO could target a single cysteine (mono-SNO) or multiple cysteines (poly-SNO), such as the case of TG2, depending on the availability of NO, the number of Cys residues, and the conformational state of target proteins [5]. Under physiological conditions, NO production is in homeostasis with a basal SNO level that keeps a subset of proteins in the resting state, such as TG2/TGase in a latent state [10]. We recognized that the SNO-RAC assay was unable to differentiate poly-SNO or mono-SNO proteins [31]. Therefore, the immunoblot showed only the changes in levels of SNO proteins eluted from thiopropyl-sepharose (Figure 6B,C).

With the availability of 3D structures of TG2 complexed with different ligands, including Ca^+2^, GDP, GTP, and Q-substrate peptidomimetics (ZDON), there were subtle differences on the surface accessible Cys residues, suggesting that there existed different conformations other than “Open” or “Close” (Table 1). The surface-exposed Cys residues were likely the initial targets for nitrosylation, but other Cys residues were not excluded due to additional protein-protein interactions. We were the first to demonstrate that poly-nitrosylated TG2 could dispense part of SNOs once ~mM of Ca^+2^ is added, indicating different conformations of TG2 had different SNO-binding capabilities [9].

An in situ TGase assay was used to investigate TG2/TGase substrates and to validate they were proximal to TG2. The BP used here was a primary amine (K-substrate), and TG2/TGase showed less specificity toward K-substrates compared to Q residues [12,15]. Selected substrates were investigated as they were either reported as TG2/TGase substrates or interacting partners and were related to redox, glycolysis, and inflammation (Figure 5B). The data demonstrated that IS treatment was sufficient in inducing intracellular Ca^+2^ levels to activate TG2/TGase. The increase in induced intracellular Ca^+2^ levels was likely to be time and IS-dosage-dependent [9]. Depending on the concentrations of IS, Ca^+2^ could be induced to various levels and with different locations and, therefore, could induce various conformations of TG2. To date, there are at least 150 TG2/TGase substrates reported on the TRANSDAB database, and TG2’s interactome in kidney tissues is also reported [12,40]. Among these, PKM2, GAPDH, β-actin, GP91-Phox, and IkBα are known TG2/TGase substrates or interacting partners [12], while eNOS, G6PD, and NFkB (P60 subunit) have not been reported (Figure 5B). Immunoblot analysis demonstrated that FN, GP91-Phox, NFkB (P60 subunit), TG2, eNOS, and IkBα were aminylated in an IS-dosage-dependent manner, and the band intensities of all substrates were reduced in the presence of ZDON, indicating ZDON was acting against TG2/TGase (Figure 5A), and with specificity. Although there existed the possibility that ZDON was acting indirectly, this would require further investigation. The use of siRNA to knock down TG2 levels resulted in poor survival of ECs. The data indicated that TG2 was indeed proximal to these substrates. Interestingly, we observed that PKM2, G6PD, GAPDH, and b-actin were aminylated by TG2/TGase even in the absence of IS, and the band intensities remained similar with increased IS concentrations, suggesting that these proteins were constitutively aminylated (Figure 5B). Physiological aminylation of substrates by TG2/TGase, especially cytoskeletal proteins, such as tubulin, has been reviewed [12].

After verifying ZDON was acting against TG2, it was used to probe TG2′s potential function as an S-nitrosylase since only substrates in close proximity could be the targets for nitrosylation. We found ZDON denitrosylated the levels of SNO-TG2 and all identified TGase substrates (except FN), including eNOS, PKM2, G6PD, NFkB, IkBa, and GAPDH, either in the presence or absence of IS (Figure 6C,D). Thus, the data suggested that S-nitrosylated ZDON-TG2 was not effectively dispensing SNOs to all its TGase substrates (Figure 6C,D), either due to conformational change or indirectly through other type of S-nitrosylase(s), and this would require further investigation. As shown in Figure 3B, higher ECARs were observed in cells pre-treated with ZDON either in the presence of IS, indicating an increase in substrate flux to the glycolytic pathway. Thus, ZDON could have an effect of increasing cellular ATP levels through glycolysis to promote survival either through TG2′s effect or other unknown mechanism(s) (Figure 1B). The data demonstrated that ZDON could regulate energy homeostasis by affecting TG2’s function, either as a TGase or as an S-nitrosylase. The possibility to function as an S-nitrosylase is discussed below.

As for the functional consequences of the differential S-nitrosylation of target proteins, eNOS undergoes a complex pattern of PTMs that regulate its activity [5]. *S*-nitrosylation of eNOS’s Cys^96^ and Cys^101^ reversibly attenuates its enzyme activity, and denitrosylation of eNOS is concomitant with its activation [5,41]. There was a decrease in SNO-eNOS together with a decrease in phospho-eNOS, demonstrating eNOS’s activity was undergoing complicated regulation at the initial phase of IS-induced injury (Figure 4B and Figure 6C,D). Although ZDON denitrosylated eNOS to enhance its activity either in the presence or absence of IS, it did not increase the P-eNOS (Figure 4B and Figure 6C,D). The results indicated that ZDON could act directly on TG2 to dispense SNOs or indirectly through other S-nitrosylase(s), and this could have an effect on vascular tone. Our results are consistent with recent studies showing that pretreatment with ZDON fails to lower vascular tone relaxation in a NO-mediated process using isolated rat-resistant arteries [42].

In addition to a role in glycolysis, GAPDH is also functioning as an S-nitrosylase [43]. Increased SNO-GAPDH is considered pro-apoptotic by nuclear translocation following binding to E3 ligase Siah1 and transnitrosylates nuclear protein DNA-PK, SIRT1, and HDAC2 [44]. S-nitrosylation of GAPDH inhibits its glycolytic enzymatic activity and may serve to regulate glycolysis [26]. In the absence of ZDON, normal cellular response to IS exposure resulted in reduced levels of SNO-GAPDH, which could shift more glucose substrate flux to PPP to defend against oxidative stress and decrease cell death (Figure 6C,D). In the presence of ZDON, there was an increase in levels of SNO-GAPDH by ~2-fold upon IS exposure, which could decrease substrate flux to the glycolytic pathway and increase cell death (Figure 6C,D). Overall, the effects of ZDON on SNO-GAPDH levels could have detrimental effects on cell survival.

PKM2 is identified as an important target for S-nitrosylation in acute and chronic inflammation, and *S*-nitrosylation of 4 of the 10 Cys residues inhibited its glycolytic activity [4,24]. An increase in SNO-PKM2 levels was reported to drive substrate flux toward the PPP for the generation of NADPH and GSH and to function as the cellular defense system against oxidative stress [24]. At 20 or 40 μM of IS, ECAR was increased in response to the reducing ATP levels, indicating more substrate flux to glycolysis to generate more ATPs (Figure 3A). However, at 80 μM of IS, ECAR was decreased to vehicle control levels indicating substrate flux was shifting to PPP to defend against oxidative stress (Figure 3B). The data were consistent with an increase in SNO-PKM2 levels associated with a concomitant decrease in PKM2 expression and an increase in G6PD expression upon exposure to 100 μM of IS (Figure 6C and Figure 7A). Thus, a biphasic response to IS-induced injury was observed. At lower levels of IS, glucose flux was shifting toward glycolysis, while at higher levels of IS, substrate flux was shifting to PPP to defend against oxidative stress as a response to the deeper loss of NADPH levels. It is unestablished whether the effects of increased SNO-G6PD on its activity. Our data support that higher SNO-G6PD levels would increase its activity toward PPP, and this would require further investigation. ZDON had an effect on the S-nitrosylation levels of PKM2 and G6PD, and the data suggested that TG2 could play a novel role either directly as an S-nitrosylase or indirectly through other S-nitrosylase(s) in regulating glycolysis and PPP.

An important transcription factor that is induced by ROS is NF*k*B, which plays a pivotal role in inflammation, cell survival, and proliferation [45]. Under physiological conditions, NF*k*B (p50/p65) is inactive due to a tight association with IkBα. In the canonical pathway of inflammation, IkBα is phosphorylated by IkB kinase (IKK), causing degradation of IkBα and translocation of NFkB to the nucleus to transactivate a variety of genes [45]. Studies reveal that NO possesses anti-inflammatory effects through the S-nitrosylation of IkBa and p50/p65 to prevent nuclear translocation of NFkB [45,46]. TG2/TGase is involved in non-canonical activation by cross-linking I*k*Bα and reducing free I*K*Bα, leading to the translocation of NF*k*B into the nucleus [47]. In another non-canonical activation pathway of NFkB, TG2 can also interact directly with IkBα leading to its degradation [48]. In the absence of ZDON, we observed increased NFkB (~1.5 fold) and slightly decreased IkBa levels at 6 h of IS (Figure 7A) that were counteracted by the increase in SNO-IkBa and SNO-NF*k*B levels to minimize inflammatory responses (Figure 6C,D). These could represent an initial cellular defense system to prevent further inflammatory response by inhibiting NFkB activation. However, in the presence of ZDON, there was an increase in NFkB expression and decreased IkBa levels upon IS exposure, suggesting an increase in inflammatory response (Figure 7A,C). These were accompanied by decreased SNO-IkBa and SNO-NF*k*B levels (Figure 6C,D), which could increase NFkB-mediated inflammatory response. Overall, the effects of ZDON could have detrimental effects by increasing NFkB-mediated inflammatory response.

Taken together, the current investigation was undertaken to understand the molecular basis for the decreased NO bioavailability on the effects of TG2/TGase and the ability of TG2 to function as an S-nitrosylase to transnitrosylate its substrates. We were focused on the proteins related to redox, glucose metabolism, and inflammatory responses at the initial phase of IS-induced EC injury. The efficacies of ZDON, a specific inhibitor of TG2/TGase, were used to probe the role of TG2 in the process.

Based on the protein expression data, the detrimental effects of IS were demonstrated by the upregulation of ACE, NFkB, and NOX2, accompanied by the downregulation of ACE2 and IkBa that could contribute to the increase in oxidative stress and inflammatory responses, leading to the uncoupling of eNOS and reduce NO bioavailability (Figure 7). ZDON pretreatment did not ameliorate the expression patterns of the aforementioned proteins; therefore, it did not significantly improve caspases activation in IS-induced injury (Figure 1C). As inhibiting TG2/TGase did not improve IS-induced injury, we investigated the role of TG2 in functioning as an S-nitrosylase. Thus, the increase in ACE and decreased ACE2 could be the upstream therapeutic targets contributing to the increased inflammation, ROS, and ECM degradations during IS-induced EC injury.

We then investigated the effects of decreased NO bioavailability on the nitrosylation of TG2 and its substrates. IS increased the S-nitrosylation levels of TG2, NFkB, IkBa, G6PD, and PKM2 despite a decrease in NO bioavailability (Figure 6B,C). The data represented a cellular defense mechanism to keep TG2/TGase in a latent state, as NO can inhibit TG2/TGase activation to minimize further injury [9,22]. Increasing S-nitrosylation of NFkB, IkBa, PKM2, and G6PD (Figure 6C,D) indicated glycolytic substrate flux was shifting to PPP to defend against ROS and to minimize cellular inflammatory responses. IS decreased the S-nitrosylation levels of eNOS and GAPDH, indicating cellular responses to activate eNOS’s activity and to minimize cell death, respectively. In the presence of ZDON, we found much lower levels of SNO-PKM2 and SNO-G6PD which could result in more glucose substrate flux to glycolysis and lower levels of SNO-NFkB and SNO-IKBa would be expected to increase cellular inflammatory response under physiological and IS-induced conditions. Based on the ZDON’s effects on S-nitrosylation, the reduced S-nitrosylation of the aforementioned proteins would not be expected to ameliorate IS-induced injury.

In summary, the current investigation suggested a novel role of TG2 in functioning as an S-nitrosylase to transnitrosylate its TGase substrates. Our data also provided novel insight into the complicated role of TG2 in cell death and survival [49]. Under normal conditions, TG2 could play a beneficial role in the nitrosylation of NFkB, IkBa, G6PD, and PKM2 to ameliorate oxidative stress and inflammatory responses in IS-induced injury (Figure 6 and Figure 8) but was insufficient to inhibit caspases activation. Although ZDON could inhibit the activation of TG2/TGase in IS-induced injury, it introduced a conformational change in TG2 that interfered either with the ability of TG2 to nitrosylate its substrate or by inhibiting the function of other S-nitrosylase(s). A different type of TG2 inhibitor that would not change the natural close conformation of TG2 may be needed to have beneficial effects and would require further investigation. Intracellular TG2 normally is in a “close” conformation, and ZDON converts it to an “open” conformation. The ZDON-bound TG2 could have altered the efficacies of transnitrosylation to its TGase substrates, as transnitrosylation is conformation- and proximity-dependent [5]. An SNO-RAC assay on purified TG2 indicated ZDON-bound TG2 contained more SNOs but could be ineffectively dispensing NO to its substrates due to the “open” conformation (Figure 6A). Future development of TG2 inhibitor(s) should take into consideration its effect on TG2′s conformational state to prevent interfering with its transnitrosylation ability. There are several irreversible peptidomimetic inhibitors developed targeting TG2, and ZED1227 is currently in a phase 2 clinical trial against liver and celiac diseases (clinicaltrials.gov, 1 June 2023). In addition, the inhibition of TG2/TGase could be used as a combinational therapy with inhibitors to the upstream ACE and AT1R blocker.

## 4. Materials and Methods

### 4.1. Chemicals

5-biotin-amido-pentylamine (BP) was obtained from Pierce (Rockford, IL, USA). Indoxyl sulfate (potassium salt) was obtained from Cayman (Ann Arbor, MI, USA). Z-DON (Z006) was from Zedira (Germany). All other reagents used in this investigation were purchased from Sigma (St. Louis, MO, USA) unless otherwise noted.

### 4.2. Antibodies

Monoclonal antibody to human TG2 (7402) was purchased from ThermoFisher, Fremont, CA, USA (clone CUB7402; Labvision ); HRP-conjugated goat vs. mouse IgG or Donkey vs. rabbit IgG were purchased from Jackson Immunoresearch (West Grove, PA, USA); all other antibodies are listed in supplementals (Appendix A).

### 4.3. SDS-PAGE and Immunoblotting

Proteins were separated either on a 9% or 10% SDS-PAGE gel before being transferred to a PVDF membrane. Immunoblots were performed using a specific primary antibody followed by a secondary antibody, and protein bands were visualized using an ECL chemiluminescent kit (ThermoFisher).

### 4.4. Cells

Primary human umbilical vein endothelial cells (HUVEC) were purchased from PromoCell (Heidelberg, Germany) and cultured in a complete endothelial cell growth medium (designated as complete EGM) containing 2% FBS and growth factors supplied by the company’s (cat # C-22010 ready to use kit). HUVEC (passage 2–4) were cultured and maintained on 10 cm culture plates (Cell-bind, Corning) until confluent and grew under standard cell culture conditions (37 °C, 5% CO_2_). Cells were split into the desired plate one day before the experiments. For Seahorse experiments, cells are attached for at least 24 h before experiments.

### 4.5. Cell Treatments and Total Lysates Preparation

In all experiments, HUVEC were grown to confluent before drug treatment. Specific conditions, including concentrations and time course, were described in detail under each figure legend. All chemicals were dissolved in DMSO and diluted directly to the desired concentrations using an experimental (or assay) medium (complete EGM was diluted 1:4 with a basal medium designated as the “assay medium”). In all experiments, the final DMSO concentration was <0.5% to prevent cell toxicity. For total lysates preparation, cells were washed quickly 3 × with PBS before resuspending the cells in an IP buffer containing 50 mM Tris, pH 7.5, 100 mM NaCl, 0.5% NP-40, 0.5% sodium deoxycholate, 2 mM EDTA, 0.5 mM PMSF, and a 1× protease inhibitor cocktail (Roche biochemicals). Cells were scraped from the wells on ice. Cells were sonicated on ice for 2 × 5 s and centrifuged at 15,000× *g* at 4 °C for 30 min to remove the insoluble debris. The soluble fractions were stored in aliquots at −80 °C and used for western blotting.

### 4.6. Cellular GSH/GSSG Assay

Cells cultured in white 96-well plates (Greiner’s white clear bottom) in an assay medium were treated with different concentrations of IS in the presence or absence of chemicals, and the GSH/GSSG ratios were determined according to the manual provided by Promega’s GSH/GSSG-Glo™ luminescent Assay kit.

### 4.7. Cellular NADP^+^/NADPH Assay

Cells cultured in white 96-well plates (Greiner’s white clear bottom) in an assay medium were treated with different concentrations of IS in the presence or absence of chemicals. The NADP^+^/NADPH ratio was determined based on the manual provided by Promega’s NADP/NADPH-Glo™ luminescent Assay kit. The reading of vehicle-treated cells was used as the control, and the values were normalized to 1.

### 4.8. XF^e^24 Seahorse Glycolysis Stress Assays

Seahorse’s XF^e^24 extracellular flux analyzer was used to analyze glucose metabolism by measuring the extracellular acidification rate (ECAR). Cells were grown in a 24-well plate (Seahorse’s XF^e^24 plate) for at least 24 h until confluent before changing to an assay medium with or without inhibitors, followed by treatment with IS. The test starts with a baseline measurement of the ECAR, designated as non-glycolytic acidification. This is followed by glucose injection to activate glycolysis, which showed an increase in the ECAR due to the formation of lactate. The cells were then challenged with oligomycin, which blocks mitochondria’s oxidative phosphorylation. Cells responded to this dramatic decrease in ATP production by activating glycolysis to its maximum level, and that results in a secondary increase in the ECAR level (designated as glycolytic reserve). The test was terminated by complete inhibition of glycolysis using the glucose analog 2-DG (2-deoxy Glucose), which returned the ECAR to its non-glycolytic level.

### 4.9. Resin Assisted Capture for S-nitrosothiols (SNO-RAC)

The analysis of SNO-modified proteins was carried out using the SNO-RAC method essentially as described [31]. Briefly, cells were directly lysed and homogenized in a HENS lysis buffer containing 100 mM Hepes/1 mM EDTA/1 mM DTPA/100 mM neocuproine (HEN), 1% SDS, 0.1% (vol/vol), Nonidet P-40, 0.2% S-methyl methanethiosulfonate (MMTS), a thiol-blocking agent, and 1× protease inhibitors (Roche) [12]. Proteins were precipitated with −20 °C acetone and re-dissolved in 1 mL of HEN/1% SDS. The precipitation of proteins was repeated with −20 °C acetone, and the final pellets were resuspended in HEN/1% SDS, and protein concentrations were determined using the bicinchoninic acid (BCA) method. Total lysates (2 mg) were incubated with freshly prepared 50 mM ascorbate/thiopropyl-sepharose (GE healthcare) and rotated end-over-end in the dark for 4 h. The bound SNO proteins were sequentially washed with HEN/1% SDS and 0.1X HEN/0.1% SDS, and total SNO proteins were eluted with 0.1 × HEN/1% SDS/10% β-mercaptoethanol (BME) and analyzed by total SNO-protein contents (as described below) and immunoblotting. For each experimental condition, three 10-cm plates were used and repeated at least twice.

### 4.10. SNO-RAC of Purified TG2

GST-TG2 was purified as described [9]. During purification, TG2 was cleaved away from GST with a factor Xa (Sigma), while GST-TG2 was bound on a glutathione (GSH)-sepharose column. After elution, TG2 was concentrated and dialyzed against 0.1 mM tris-acetate, pH 7.5, and 0.1 M EDTA and stored at −80 °C. Purified TG2 was nitrosylated with a freshly made CysNO donor (TG2:CysNO = 1:200) at RT for 10 min, and SNO-TG2 was identified using the SNO-RAC described above.

### 4.11. Total SNO Proteins Content

Three 10-cm plate confluent HUVECs for each experimental condition were harvested to perform an SNO-RAC assay, and proteins eluted from isopropyl-sepharose were dialyzed against 2 × 2 L of Tris-Cl, pH 8.0, 100 mM NaCl, 2 mM EDTA, and 0.5 mM DTT to remove BME before determining protein concentrations by a BCA assay (Pierce, ThermoFisher).

### 4.12. In Situ TG2/TGase Assay

Cells were pre-incubated with 0.5 mM of BP for 1 h before being treated with different concentrations of IS in the presence or absence of inhibitors for an additional 5 h. Total lysates were prepared as described earlier. The sample was loaded on a 9% SDS-PAGE and processed for immunoblotting and then probed with streptavidin-HRP antibody or other antibodies. Bands were developed using Pierce’s chemiluminescence kit.

### 4.13. Intracellular ROS Generation Assay

Cells were cultured in an EGM medium until confluent in a 24-well plate. After washing 1× with PBS followed by incubating with either 10 μM of 2′,7′-dichlorofluorescein diacetate (DCFH-DA) (Molecular probe) or 2 μM of Mitosox (to visualize mitochondria’s ROS) in a basal medium for at least 30 min at 37 °C. After removing DCFH-DA (or Mitosox) solutions, cells were incubated with different concentrations of IS (with and without chemicals) at 37 °C for different time intervals in the dark. Fluorescence was visualized using a fluorescent microscope equipped with a Fitc or red filter.

### 4.14. Molecular Modeling

The three-dimensional (3D) structures of TG2 complexed with Ca^+2^ (pdb:6kzb), GTP (pdb: 4PYG), GDP (pdb: 1kv3), and ZDON (pdb: 3S3J) are available from RCSB Protein 3D Data Bank (rcsb.structure) and the online “3D view” tools were used to modeling the surface accessible Cys residues of TG2.

### 4.15. Statistical Analysis

Data were analyzed using the ANOVA test, followed by Bonferroni’s test and Student’s *t*-test when appropriate (SPSS for Windows, version 20.0 (IBMCorp., Armonk, NY, USA)). *p* < 0.05 was considered statistically significant.

## Figures and Tables

**Figure 1 ijms-24-10935-f001:**
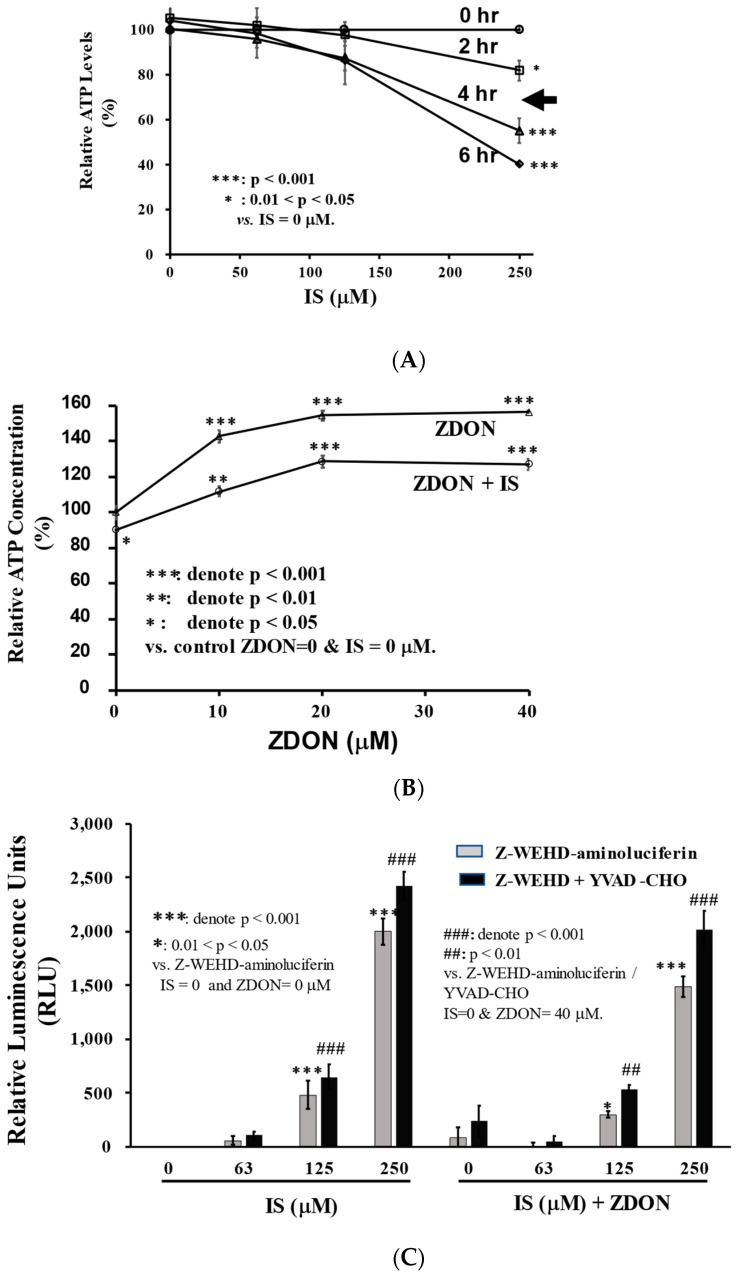
Effects of IS and ZDON on cellular ATP levels and apoptotic response. (**A**) Time course of IS on cell’s ATP levels. Cells grown in 96-well white plates in an assay medium were treated with 0 to 250 μM of IS for 0, 2, 4, or 6 h, followed by measuring ATP levels using Promega’s CellTiter-Glo Luminescent Cell Viability kit (to measure ATP). (**B**) Cells were either treated with ZDON (0 to 40 μM) alone for 4 h or pre-treated for 4 h with ZDON (0 to 40 μM), followed by 4 h of incubation with 100 μM of IS. Cellular ATP levels were measured using the same kit as described in 1A. (**C**) Effects of IS and ZDON on the activation of caspases. Cells in triplicate wells grown in a white 96-well plate in assay medium were pre-treated 2 h either with or without ZDON (40 μM) followed by 4 h of incubation with 0, 63, 125, or 250 μM of IS for 4 h. Caspase 1 activities were then measured using Promega’s Caspase-Glo^®^ 1 Inflammasome Luminescent Assay. For the development of luminescence, caspase-1 substrate, Z-WEHD-aminoluciferin was used either in the presence or absence of 1 μM of Ac-YVAD-CHO, a specific caspase-1 inhibitor. The proteasome inhibitor, MG-132, was also included in the reagent to eliminate nonspecific proteasome-mediated cleavage of the substrate. Triplicate wells were used for each concentration, and results are presented as means ± standard deviation (SD). Each value was normalized to the vehicle-treated cells and is presented as a percentage (**A**,**B**).

**Figure 2 ijms-24-10935-f002:**
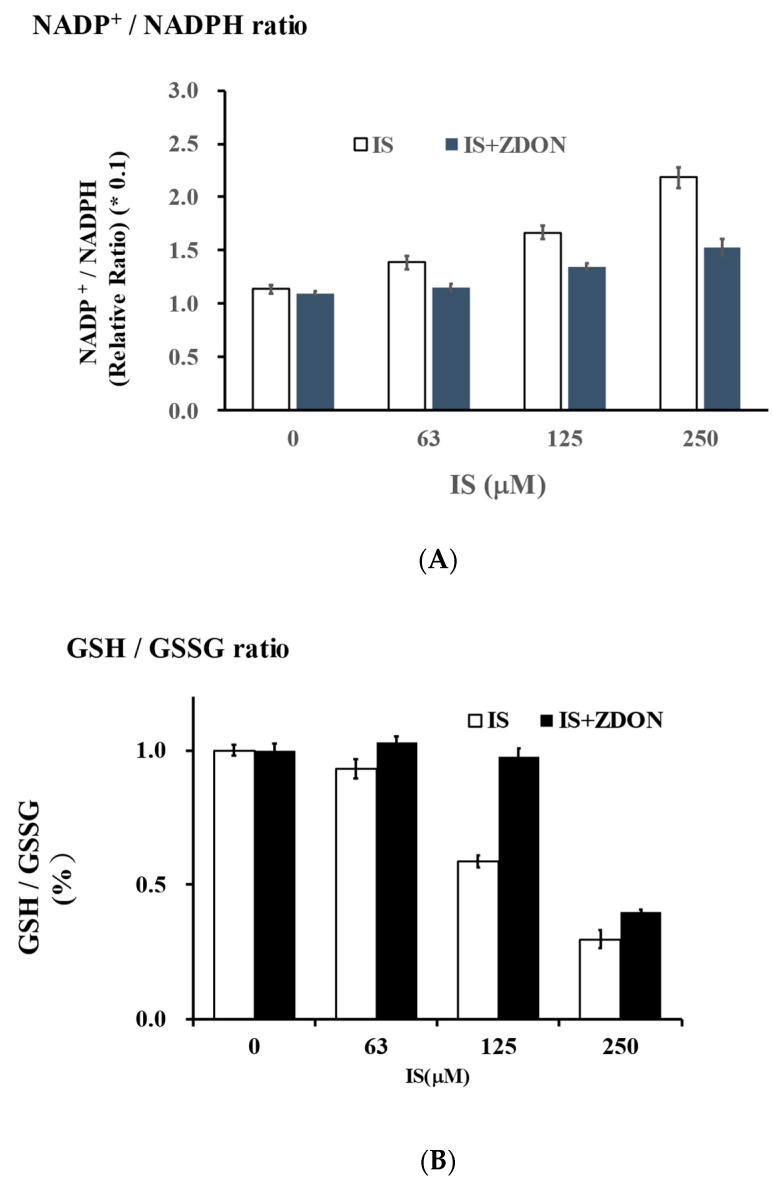
Effects of IS on NADP^+/^NADPH and GSH/GSSG ratios in cells previously treated either with or without ZDON. In (**A**,**B**), the NADP^+^/NADPH and GSH/GSSG ratios are shown, respectively. Cells were grown in 96-well white plates pre-treated with either the vehicle control or ZDON (40 μM) for ON, followed by 4 h of incubation with 0–250 μM of IS in the assay medium. Triplicate wells were used for each concentration. After IS treatment, the NADP^+^/NADPH or GSH/GSSG ratios were determined using Promega’s NADP^+^/NADPH-Glo™ or GSH/GSSG-Glo™ Assay kits, respectively. The vehicle-treated cells were used as the control, and the values were normalized to 1. Data are presented as means ± SD.

**Figure 3 ijms-24-10935-f003:**
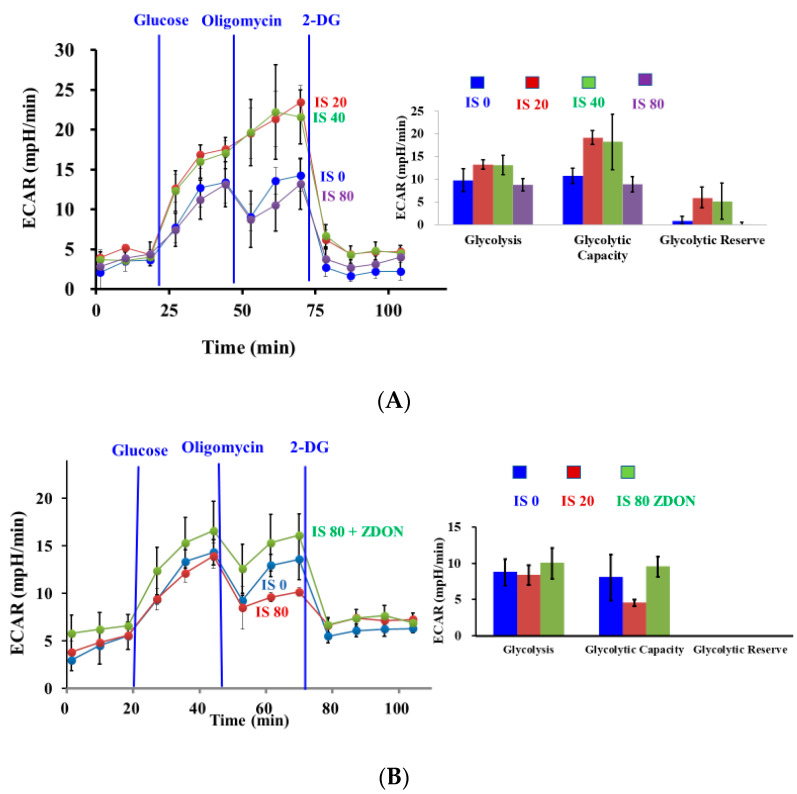
Effects of ZDON and IS on glycolysis stress. (**A**) Effect of IS on extracellular acidification rate (ECAR). Cells were grown in 24-well plates (Agilent Inc.) in triplicate, followed by incubation with 0, 20, 40, and 80 μM of IS for 3 h in an assay medium before transferring to a Seahorse XF-24 instrument for the measurement of ECAR. (**B**) Effects of ZDON on ECAR. Cells were pre-treated either with or without ZDON for 1 h, followed by incubation with 0 or 80 μM of IS for 3 h in assay medium before transferring to a Seahorse XF-24 instrument for the measurement of ECAR. At least triplicate wells were performed for each concentration. Glycolysis stress was performed by acute injection of glucose, oligomycin, and 2-DG according to instruction manuals. The data on glycolysis, glycolytic capacity, and glycolytic reserve are also indicated.

**Figure 4 ijms-24-10935-f004:**
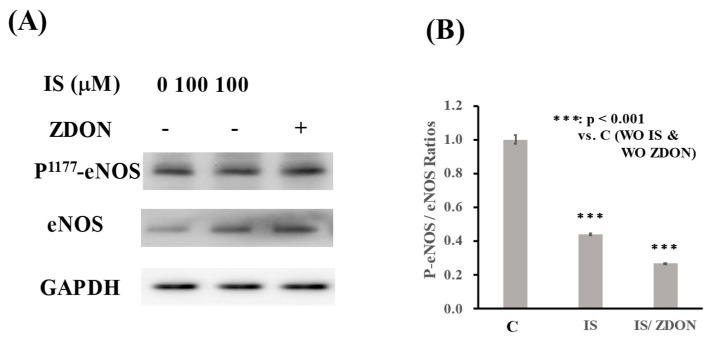
Expression of eNOS and phospho-eNOS in cells pre-treated with either ZDON followed by incubation with IS. (**A**) After previously being treated with ZDON (40 μM) for 1 h, cells were incubated with either the vehicle control or 100 μM of IS for 4 h. Total soluble lysates (20 μg) from each sample were loaded on a 9% SDS-PAGE and processed for immunoblotting with either eNOS or P^1177^- eNOS specific antibodies. Bands were visualized using Pierce’s chemiluminescence kit. (**A**) The intensities of each band in (**A**) were quantified using the Image J program and are presented in (**B**). The intensities were presented as means ± SD and are plotted as P^1177^-NOS / eNOS. *** denotes *p* < 0.001.

**Figure 5 ijms-24-10935-f005:**
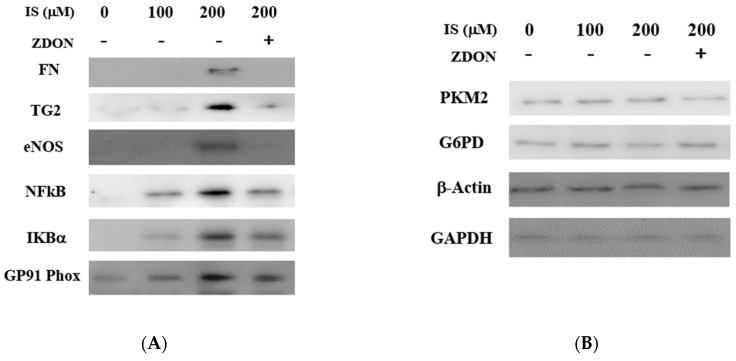
Immunoblotting of in situ TG2/TGase substrates in cells previously treated either with or without IS. (**A**,**B**) Cells were pre-incubated with 0.5 mM of BP either with vehicle control or ZDON (40 μM), followed by incubation with either 0, 100, or 200 μM of IS for 5 h and isolation of total lysates as described under Section 4. After the pull-down of total biotin-labeled proteins using streptavidin beads, 10 mL of the samples eluted fractions from streptavidin beads were separated by a reducing 9% SDS-PAGE and processed for immunoblotting and probed with individual specific antibodies. Bands were visualized using Pierce’s chemiluminescence kit.

**Figure 6 ijms-24-10935-f006:**
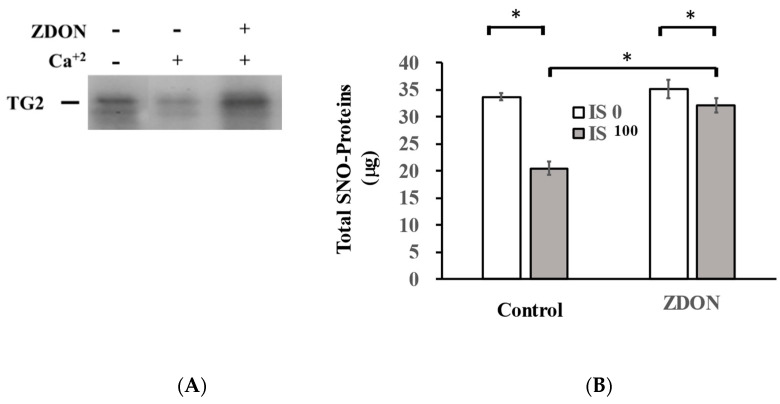
Determination of total SNO proteins and specific SNO proteins in cells pre-treated either with or without ZDON, followed by incubation with IS. (**A**) SNO-RAC on purified TG2. Purified recombinant TG2 was preincubated either with 0, 1 mM Ca^+2,^, or 1 mM Ca^+2^/40 μM ZDON on ice for 5 min before nitrosylated with CysNO at RT for 10 min as described under *Materials and Methods*. Then, SNO-TG2 was identified using an SNO-RAC assay. (**B**) Total SNO-protein contents. After being pre-treated either with the vehicle control or ZDON (40 μM) for ON, followed by 5 h incubation with 100 μM IS, cells were processed with an SNO-RAC assay as described under Section 4. Total SNO proteins eluted from thiopropyl-sepharose beads were quantified using a protein BCA assay as described under Section 4. * denotes *p* < 0.05. (**C**) Specific SNO proteins. Equal amounts of eluted proteins from (**B**) were analyzed by immunoblots. (**D**) The protein band’s intensities in Figure 6C were quantified using the Image J program and are shown in Figure 6D and are presented as means ± SD.

**Figure 7 ijms-24-10935-f007:**
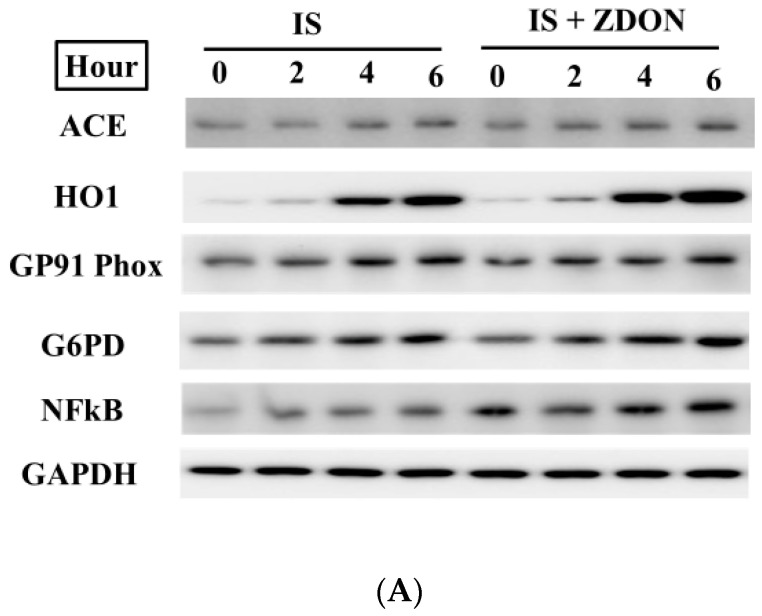
Up- and down-regulated proteins upon exposure to IS. (**A**) Expression of up-regulated proteins. After treatment with either the vehicle control or ZDON (40 μM), followed by incubation with 100 μM of IS in an assay medium for either 0, 2, 4, or 6 h, total cellular lysates were prepared in an IP buffer as described under Materials and Methods. Depending on the expression levels, the same amount of lysates was loaded on an SDS-PAGE, followed by immunoblotting. Different amounts of lysates for each specific protein were loaded to ensure luminescent signals were in the linear range. The expression of ACE, HO1, GP91/Phox, NFkB, and G6PD are shown. (**C**) Expression of down-regulated proteins. Cells were treated as described in (**A**). The expression of ACE2, FN, IkBa, and PKM2 are shown. In all samples, GAPDH was used as the loading control. The intensities of each protein band in (**A**,**C**) were scanned and quantified using the Image J program and are presented as means ± SD either in (**B**,**D**), respectively.

**Figure 8 ijms-24-10935-f008:**
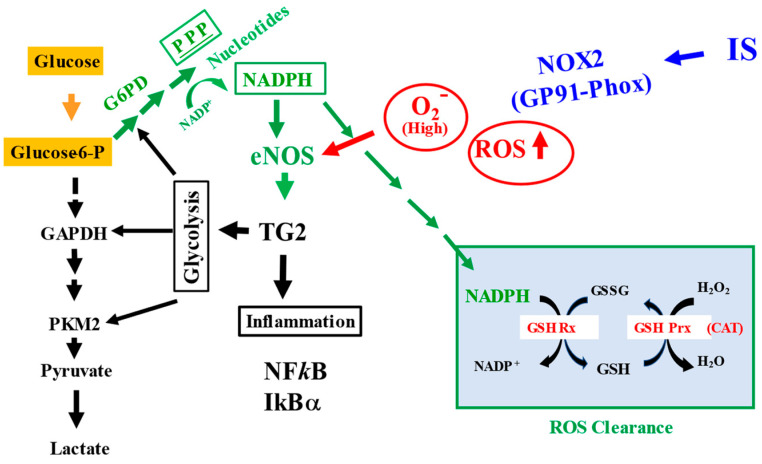
Interplay of TGase and S-nitrosylase of TG2 under normal and IS-induced injury. In endothelial cells, glycolysis and PPP are major pathways for maintaining the homeostasis of energy and oxidative stress and supplying sufficient NADPH for eNOS and ROS clearance. Oxidative stress induced by IS has harmful effects by the uncoupling of eNOS and the interplay of TGase and S-nitrosylase functions of TG2. ACE, angiotensin-converting enzyme; GSH Rx, glutathione reductase; GSH Prx, Glutathione Peroxidase; CAT, catalase; GP91-Phox.

**Table 1 ijms-24-10935-t001:** Accessible cysteine residues in TG2 complex with different ligands.

Ligand	PDB Code	Cysteine Number
GDP	1KV3	98, 545, 554
GTP	4PYG	10, 98, 269, 545, 554
Ac-P(DON)LPF-NH2	2Q3Z	10, 98, 269, 545, 554
Z-DON-VPL-Ome (ZDON)	3S3J	10, 98, 545
Ca^2+^	6KZB	10, 27, 98, 269, 545, 554

Accessible cysteine residues were modeled using the “3D view” tool available on the protein 3D structure website (rcsb.structure, 1 June 2023).

## Data Availability

The datasets generated during and/or analyzed during the current study are available either in Appendix A and/or from the corresponding author upon reasonable request.

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
