# Peer review of "S-Nitrosylation of Tissue Transglutaminase in Modulating Glycolysis, Oxidative Stress, and Inflammatory Responses in Normal and Indoxyl-Sulfate-Induced Endothelial Cells"

_ijms, 2023, doi:10.3390/ijms241310935_

Round 1

Reviewer 1 Report

Dear authors,

Please read the manuscript carefully as it has numerous typos, such as 'mediumwith'; 'mediumdesignated' etc.  I would also recommend revising/editing the following sentences:

1) Cardiovascular disease (CVD) occurs early and declining renal function is associated with cardiovascular morbidity and mortality in CKD patients (line 41).

2) However, the regulation of ACE and its beneficial isoform ACE2 in IS-induced EX is largely unknown (line 48).

3) The production of NO during eNOS uncoupling reacting with O2 - can form cytotoxic peroxynitrite (ONOO-), which can contribute to oxidation of the BH4 (line 64).

My other suggestions include combining Figures 1B and 1C together for better visualization. The figure 3B seems to miss data on glycolytic reserve. 

I would recommend minor English editing to improve clarity and eliminate errors and typos.

Reviewer 2 Report

The manuscript “S-nitrosylation of tissue transglutaminase in modulating glycolysis, oxidative stress, and inflammatory responses in normal and indoxyl sulfate induced endothelial cells.” by Cheng-Jui Lin et al describes nitrosylation of tissue transglutaminase in regulating endothelial cell function in response to the circulating uremic toxin indoxyl sulfate. The investigators use primary, although expanded, human umbilical cells and biochemical and cell biologic assays to find indoxyl sulfate inhibits endothelial cell function and viability and that addition of a tissue transglutaminase inhibitor ameliorates many of the resulting dysfunctions. Overall, this is an interesting topic and mechanism, but poor execution of the experimental plan prevents drawing the proposed conclusions.

Major:

1. As noted, this is a relevant and important topic that uses appropriate approaches. However, use of ZDON at excessive levels, a lack of consistency in choice of indoxyl sulfate or ZDON concentrations, lack of all controls, and poor presentation greatly impact the manuscript.

2. The level of indoxyl sulfate used here (250 uM) exceeds at least early identification of indoxyl sulfate levels associating with kidney disease ( Barreto et al Clin J Am Soc Neprol 4:1551, 2009). This needs to be addressed in light of Fig 1A concentration response results. Might a longer preincubation increase intracellular levels in cultured HUVEC?

3. All panels in Fig 1 require range/errors, and statistical comparisons.

4. Text describing Fig. 1 states different caspases are affected, but in fact are not shown. 

5. Fig. 1D requires ZDON alone as a control.

6. The IC50 for ZDON is stated as 20 nM, but ZDON is used at 2000-times this level. Off targets effects may result at this level.

7. Fig. 1 shows >100 uM indoxyl sulfate suppresses HUVEC ATP, but Fig. 2 A and B use 100uM, and often 125 and 250 uM.

8. Figure 2 does not present the effect of ZDON along as a key control.

9. Fig. 3B Seahorse tracing show alterations at 20 and 40 uM indoxyl sulfate, with 80 uM giving the same response as buffer. However, Fig. 3b now uses 80 uM indoxl sulfate ± ZDON. 

10.  Fig. 4A western blot shows little difference, and does not support Fig. 4B compilation.

11. Fig. 5 shows no effect of indoxyl sulfate at the 100 uM level shown as the maximal concentration that does not suppress HUVEC ATP levels.

12. Fig. 6 shows ZDON, only in the presence of ascorbate, greatly enhances SNOylation of key proteins that is independent of indoxyl sulfate. This is significant. However, L341-345 state this is an effect of indoxyl sulfate, but this is incorrect: it is an effect of ascorbate and not indoxyl sulfate.

13. Fig. 7 does not present useful information or differences among conditions.

Editorial

1) Reference 1 is not appropriate and does not support its stated role here.

2) L92 run on

3) L104, break into new paragraph

4) the introduction is far too excessive. Some should be moved to the Discussion

5) But, the Discussion also is far too detailed.

6) ATP concentration in Fig 1A is not equivalent to viability!

8) Fig 1B and C should be combined

9) Labeling of the figure is incorrect and misleading. Z-WEHD was not used as stated, but rather a fluorogenic derivative as the assay substrate. YVAD is then used as a caspase 1 inhibitor.

Round 2

Reviewer 2 Report

Presentation in this version is improved and more precise. Fig. 1 is now easily understandable and statistical comparisons are now (except for some reason Fig. 2 where the differences will be significant) presented.

Fig. 6 however is not presented optimally. Visual inspection shows ascorbate is the primary difference. The authors note that the ascorbate is required for the assay but the lack of ascorbate control then might be stated, but is highly distracting for this figure.